# Role of Protein Tyrosine Phosphatases in Inflammatory Bowel Disease, Celiac Disease and Diabetes: Focus on the Intestinal Mucosa

**DOI:** 10.3390/cells13231981

**Published:** 2024-11-29

**Authors:** Claudia Bellomo, Francesca Furone, Roberta Rotondo, Ilaria Ciscognetti, Martina Carpinelli, Martina Nicoletti, Genoveffa D’Aniello, Leandra Sepe, Maria Vittoria Barone, Merlin Nanayakkara

**Affiliations:** 1Department of Translational Medical Science, Section of Pediatrics, University Federico II, Via S. Pansini 5, 80131 Naples, Italy; claudia.bellomo@unina.it (C.B.); francesca.furone@gmail.com (F.F.); ilariaciscognetti@gmail.com (I.C.); martinacarpinelli@gmail.com (M.C.); martina.nicoletti0@gmail.com (M.N.); 2Department of Medicine and Surgery, University of Parma, 43121 Parma, Italy; robertarot.97@gmail.com; 3ELFID (European Laboratory for the Investigation of Food-Induced Diseases), University Federico II, Via S. Pansini 5, 80131 Naples, Italy; jedaniello1998@gmail.com; 4Department of Molecular Medicine and Medical Biotechnology, University of Naples Federico II, 80131 Naples, Italy; leandra.sepe@unina.it; 5Federico II University Hospital, 80131 Naples, Italy; merlin.nanayakka@unina.it

**Keywords:** protein tyrosine phosphatases, celiac disease, diabetes, inflammatory bowel diseases, intestinal mucosa

## Abstract

Protein tyrosine phosphatases (PTPs) are a family of enzymes essential for numerous cellular processes, such as cell growth, inflammation, differentiation, immune-mediated responses and oncogenic transformation. The aim of this review is to review the literature concerning the role of several PTPs—PTPN22, PTPN2, PTPN6, PTPN11, PTPσ, DUSP2, DUSP6 and PTPRK—at the level of the intestinal mucosa in inflammatory bowel disease (IBD), celiac disease (CeD) and type 1 diabetes (T1D) in both in vitro and in vivo models. The results revealed shared features, at the level of the intestinal mucosa, between these diseases characterized by alterations of different biological processes, such as proliferation, autoimmunity, cell death, autophagy and inflammation. PTPs are now actively studied to develop new drugs. Also considering the availability of organoids as models to test new drugs in personalized ways, it is very likely that soon these proteins will be the targets of useful drugs.

## 1. Introduction

Protein tyrosine phosphatases (PTPs) are enzymes that remove phosphate groups from tyrosine residues on other proteins. Tyrosine kinases are, on the other hand, enzymes that add phosphates to tyrosine residues. Owing to the activity carried out by protein kinases and phosphatases, the regulation of phospho-tyrosine signalling is possible, so the cells are ready to respond to external signals, inducing changes in the intracellular environment. When receptors on the cell surface bind to ligands, this results in the transduction of external signals within the cell. Therefore, RTKs (receptor tyrosine kinases) can dimerise upon ligand binding, leading to trans autophosphorylation of the receptor. This event leads to the recruitment of proteins that bind phospho-tyrosine residues to the receptor, which propagates and amplifies a variety of signal cascades [1]. The kinases and phosphatases have complementary functions in regulating cell signalling with the kinases controlling the amplitude of a signalling response and the phosphatases controlling the rate and duration of the response. PTPs are a super family with 100 circa different genes while the kinases are circa 90 genes, indicating the same level of complexity for the two families [2]. Moreover, they reach a higher level of complexity considering post-transcriptional modifications, alternative promoters or alternative splicing. Such diversity, on the one hand, makes a neat classification hard; on the other hand, it points to a wide range of biological functions that are not completely clarified. Understanding the “phosphatome”, is currently a real challenge to understand the specificity and roles of all phosphatases. In fact, it is not possible to generalize the function of each phosphatase: some of them are very selective, whereas others are more promiscuous [3]. Therefore, PTPs are signalling molecules that regulate a variety of cellular processes, such as cell growth, differentiation, the cell cycle, oncogenic transformation, embryonic development, immune response, inflammation, proliferation, differentiation, cell adhesion and autophagy [4,5,6,7].

PTP1B was the first phosphatase identified. This protein has a sequence analogous to a segment of the receptor protein CD45 (Cluster of Differentiation 45) [8]. Owing to this structural analogy, it is possible to identify the existence of a conserved catalytic domain, with a cysteine amino acid residue, that has become the main feature of the PTP gene family [9].

Members of the PTP family are classified according to their structural and biochemical properties.

Human PTPs are classified into four categories depending on the substrate specificity and amino acid sequence. Class 1 consists of the classical PTPs, including the receptor-type protein tyrosine phosphatases (RPTPs) and non-receptor-type protein tyrosine phosphatases (nRPTPs). Class 2 comprises the dual-specificity phosphatases (DSPs) that target phosphorylated tyrosine, serine and threonine residues. The low-molecular-weight phosphatases (LMW-PTPs) belong to class 3 and the CDC25 class phosphatases form class 4 [10] (Figure 1). At the cellular level, the PTP proteins are mainly cytosolic, nucleo-cytosolic and receptor proteins. Although, at the tissue level, they are expressed mainly in the developing embryo, the immune system and the nervous system, they are also expressed at the level of the intestine [11].

The importance of tyrosine phosphorylation in normal cell physiology is highlighted by the fact that many human diseases are the result of aberrant protein tyrosine kinase (PTK) or protein tyrosine phosphatase (PTP) function [12]. PTP mutations that cause hereditary disease phenotypes have been described to date for T-cell Leukemia, Rheumatoid arthritis, Hashimoto thyroiditis, Addison Disease, Noonan and Leopard syndrome and others [13]. Interestingly, both loss-of-function and gain-of-function mutations in these genes may underlie the reported disease. Most studies in the literature have described both the molecular mechanisms and the therapeutic target of PTPs in cancer [14,15].

Moreover, the role of PTPs in inflammatory diseases such as inflammatory bowel disease (IBD), celiac disease (CeD) and type 1 diabetes (T1D) has been established [9,16,17,18].

IBD, CeD and T1D share genetic alterations of genes involved mainly in the pathways of intestinal barrier dysfunction, immune/inflammatory pathways and autophagy [19,20]. A notable aspect of these three diseases is the presence of increased intestinal permeability that precedes the onset of inflammation [21]. Moreover, both T1D and IBD patients have an increased risk of CeD [19].

IBD and CeD share a specific target organ, the intestine. Both are complex diseases in which genetics and the environment contribute to the dysregulation of innate and adaptive immune responses, leading to chronic inflammation and disease. T1D has as an organ target not only pancreatic β-cells but also the intestinal mucosa. In fact, clinical and experimental research has revealed that diabetes mellitus is characterized by intestinal hypomotility, gut microbial dysbiosis, increased gut permeability and dysfunction of intestinal stem cells, which may be linked to inflammation of the intestinal mucosa [22,23,24].

All previously described conditions are characterized by the presence of genetic and environmental factors and inflammation. Among the environmental factors that trigger inflammation, nutrients play a significant role in inducing low-grade inflammation, which is considered a “common background” for various diseases, including diabetes, IBD and CeD [25]. A new concept emerging in the recent literature suggests that genetic predisposition can increase the susceptibility of cells to inflammatory triggers, leading to cellular vulnerability and autoimmunity [25,26]. How this cellular vulnerability is characterized is not yet clear, but signalling molecules, including PTPs, are emerging as possible culprits.

The aim of this review is to review the recent literature on the role of PTPs at the level of the intestine in inflammatory diseases, such as CeD, IBD and T1D.

## 2. Materials and Methods

We selected the most recent literature on IBD, T1D and CeD, which are correlated with/associated with PTP proteins. Citing 72/103 manuscripts from the last 20 years, 51/103 are from the last 10 years, 36/103 are from the last 5 years and only 3 are from between 1993 and 1998 as they are the original references for the PTP described.

The software used to create the figures were Clustal, Microsoft Power Point and BioRender. In particular, Clustal was used to create Figure 1; BioRender was used for Figure 2; and Microsoft Power Point was used for the graphical abstract, Figure 3.

## 3. PTPs and Inflammatory Bowel Disease

In this section, we describe the role of the PTPs involved in the development of IBDs (Crohn’s disease (CD) and ulcerative colitis (UC)) at the level of the intestine, namely PTPN22, PTPN2, PTPN6, PTPN11, PTPσ DUSP2, and DUSP6 (Figure 2) (Table 1).

PTPN22 (Protein tyrosine phosphatase non-receptor 22) is expressed mainly in haematopoietic cells and is associated with multiple inflammatory disorders [1,9,27]. A gene association study demonstrated that SNPs generating the loss-of-function variant in PTPN22 were associated, respectively, with CD and UC [28,29,30]. In mouse models of experimental arthritis and DSS (dextran sulphate sodium)-induced colitis, PTPN22 deficiency resulted in increased intestinal susceptibility to inflammatory agents and pronounced disease progression, indicating that PTPN22 is required to protect individuals from systemic and gastrointestinal inflammation. Katakura et al. [31] and Yarilina et al. [32] reported that, in mice, PTPN22 promotes intestinal inflammation suppression in models of colitis and can ameliorate intestinal mucosa injury [33]. In mouse models, PTPN22 represses acute inflammatory immune responses in the gut in the presence of IL-10 (interleukin-10) but promotes acute gut inflammation in the absence of IL-10. During chronic intestinal inflammation, however, PTPN22 promotes gut inflammation irrespective of the presence of IL-10 [34]. In addition, PTPN22 at the level of the intestine is involved in autophagy and inflammatory regulation through the dephosphorylation of NOD-like receptor protein 3 (NLRP3) to allow for efficient inflammasome activation. The loss of functional autophagy counteracts the reduction in NLRP3 activation observed in PTPN22-deficient cells, leading to increased IL-1β secretion [35,36]. Defects in autophagy result in impaired goblet cell function, and variants in genes involved in autophagy are associated with an increased risk of developing IBD [37] (Figure 2 and Table 1).

The PTPN2 protein has the highest expression levels in lymphoid tissue. Previous studies in mice have provided insight into the crucial role of PTPN2 in intestinal homeostasis, including the regulation of pro-inflammatory cytokine secretion, autophagy, intestinal epithelial barrier function and gut microbiota homeostasis [38,39]. The anti-inflammatory role of PTPN2 is highlighted by the fact that PTPN2-deficient mice die a few weeks after birth because of systemic inflammation and severe colitis [40]. PTPN2 crucially contributes to normal macrophage‒IEC (intestinal epithelial cell) interactions that govern macrophage differentiation and epithelial barrier function and represents a target for therapeutic intervention in IBD. PTPN2 has also been shown to play an important role in Paneth cell regulation. PTPN2 knockout (KO) mice presented a reduced number of Paneth cells, increased Paneth cell endoplasmic reticulum stress and dysbiosis [38,41].

Clinically relevant loss-of-function mutations in the PTPN2 gene contribute to the etiopathogenesis of chronic inflammatory intestinal disease, at least in part through the dysregulation of epithelial barrier properties. As well as PTPN22, PTPN2 was also shown to regulate inflammasome activation, IL-1B secretion and autophagy regulation [40]. Consistent with these findings, PTPN2 levels were increased in the intestinal epithelium in active CD patients. [21,42,43]. In fact, PTPN2 protects the intestinal barrier by restricting the capacity of INF gamma to increase epithelial permeability and TNFα-induced signalling and cytokine secretion in human intestinal epithelial cells [21,42,43,44] (Figure 2 and Table 1).

PTPN11 was upregulated in both UC patients and a colitis model in mice [45,46,47]. It is a cytoplasmic protein tyrosine phosphatase that functions as a positive regulator of the Ras–mitogen-activated protein kinase (MAPK) signalling pathway downstream of various growth factors and cytokines, contributing to the regulation of cell proliferation and differentiation. Yamashita et al. demonstrated that mice lacking PTPN11 specifically in IECs developed severe colitis and alterations in the intestinal mucosa, with the numbers of absorptive enterocytes and goblet cells markedly reduced in the small intestine and colon with respect to controls [48]. Abnormal development of the intestinal epithelium and disrupted host–microbiota equilibrium was also described in PTPN11-deficient mice [49,50]. Moreover, Xiao et al. demonstrated that PTPN11 expressed in the monocyte/macrophage lineage serves as a pro-inflammatory factor in the intestine [51]. However, the precise role of PTPN11 in the regulation of gastrointestinal stem cell and progenitor functions, especially in vivo, has remained unclear (Figure 2 and Table 1).

PTPσ (protein tyrosine phosphatase receptor S) consists of a cell adhesion molecule-like domain, a transmembrane domain and a cytosolic region. PTPσ expression is developmentally regulated and found primarily in the nervous system and specific epithelia. PTPσ −/− mice presented high neonatal mortality. Analysis of the intestinal tissue of surviving mice revealed the presence of mucosal inflammation, intestinal crypt branching and villus blunting associated with human IBD. Additionally, PTPσ targets apical junction complex proteins in the intestine and regulates epithelial permeability [52,53] (Figure 2 and Table 1).

PTPN6 is predominantly expressed in hematopoietic and epithelial cells. It was associated with increased proliferation and altered goblet and Paneth cell differentiation. Previous studies demonstrate that the levels of PTPN6 were reduced in active UC and CD with respect to controls [54]. In another study, two PTPN6 SNPs were genotyped in a population of 107 IBD patients from Southern Tunisia and a weak association with UC was identified [55]. For this reason, a link between PTPN6 and IBD still needs to be defined (Figure 2 and Table 1).

DUSP2 was first identified as a mitogen-inducible gene in human T cells. DUSP2 is a nuclear protein phosphatase regulated by extracellular signal-regulated kinase 1/2 (ERK1/2) and p38 MAPKs. DUSP2 is a positive mediator of inflammation [56]. DUSP2 −/− mice exhibit increased mucosal hyperaemia, increased colonic ulceration and high levels of pro-inflammatory cytokines, including IL-6, IL-17, Tumour necrosis factor α (TNFα) and IL-1β. In a study of UC patients, the level of blood DUSP2 mRNA expression was reduced [57] (Figure 2 and Table 1).

DUSP6 is a cytoplasmic phosphatase specific to the ERK protein kinase [58]. The signal transduction pathway regulated by ERK is responsible for the regulation of multiple cellular responses. DUSP6 KO mice have shown CD4+ T cell polarization; altered differentiation of T-helper 1 (TH1), TH2 and TH17 cells; and an increased rate of cell proliferation and of cell death. This depletion also causes severe inflammation in mouse models of IBD with intestinal epithelial hyperplasia, the depletion of caliciform cells and infiltration of mononuclear cells. Furthermore, colon explants secrete increased levels of TNF-α and Interferon-γ (IFN-γ). In these mice, ERK inhibition significantly reduces colitis. Additionally, Beaudry K. et al., using ex vivo DUSP6 KO organoid cultures, demonstrated that the loss of DUSP6 caused an increase in crypt depth and epithelial cell proliferation [59] (Figure 2 and Table 1).

In conclusion, PTPs have a nodal role in IBD pathogenesis. It is notable that they play an important role in the intestinal epithelium, mostly regulating its homeostasis. This indicates that the intestinal epithelium compartment could be the key both to the understanding the of the molecular pathways responsible for the mucosal alterations and to find new targets for therapeutic intervention. In this context, it is important to highlight the role of intestinal organoids derived from staminal cells that represent the intestinal epithelium not only of a specific disease, but also of a specific patient, opening the door for precision medicine in these diseases.

**Table 1 cells-13-01981-t001:** PTP proteins involved in inflammatory intestinal diseases and their function at the level of the intestine investigated in several study models.

PTP Protein	Disease	Alteration in the Disease	Models of Study	Function at the Level of the Intestine	Reference
PTPN22	CD-UC	Loss of function	Mice [31,32,33,34]; Cell culture [35,36,37]	(1) PTPN22 has a dual context-dependent role: -in acute gut inflammation, PTPN22 together with IL-10 is a repressor of inflammtory response; PTPN22 minus IL-10 is a promoter -during chronic inflammation, PTPN22 promotes gut inflammation irrespective of the presence of IL-10 (2) protections from systemic and gastrointestinal inflammation (3) autophagy regulation (4) inflammasoma regulation	[1,9,27,28,29,30,31,32,33,34,35,36,37]
PTPN2	CD-UC	Loss of function	Mice [38,39,40,41] Intestinal epithelial cell (IEC) lines (T84;HT29cl.19a) [42,44]	INTESTINAL HOMEOSTASIS -regulation of pro-inflammatory cytokine secretion -autophagy -intestinal epithelial barrier function -Paneth cell regulation -autophagy regulation -inflammasoma activation -gut microbiota homeostasis	[21,38,39,40,41,42,43,44]
PTPN2	Diabetes	Minor allele frequency	Patients	Involved in many leaky gut-associated diseases	[60,61,62]
PTPN11	UC	Up-regulated and Down-regulated	Mice [45,46,47,48,49,50] Cell culture [51]	Homeostasis of IECs of absorptive enterocytes and goblet cells; microbiota equilibrium; positive regulator of MAPK signalling pathway; regulation of cell proliferation and differentiation	[45,46,47,48,49,50,51]
PTPσ	UC	Down-regulated	Mice [52,53]	PTPσ is a regulator of mucosal inflammation, intestinal epithelial development and permeability	[52,53]
PTPN6	UC	Down-regulated	Patients	PTPN6 is associated with increased proliferation and altered goblet and Paneth cell differentiation	[54,55]
DUSP2	UC	Low levels (mRNA)	Mice [46]	DUSP2 is a regulator of mucosal inflammation and colonic ulceration	[56,57]
DUSP6	UC	Down-regulated	Mice [47] HT29 and HCT116 colorectal carcinoma cells [47]	DUSP6 is a modulator of CD4+ T cell activation and epithelial cell proliferation	[58,59]
PTPRK	CeD	Down-regulated	Intestinal organoids [9]	PTPRK is a regulator of epithelial inflammation and enterocyte proliferation	[18,63,64,65,66,67,68,69,70,71,72,73,74,75,76,77]

## 4. PTPs and Diabetes

T1D is an autoimmune disease characterized by the destruction of insulin-producing β-cells in the pancreas. The onset of T1D is thought to be due to a complex interplay of genetic and environmental factors. The primary genetic risk factors include specific human leukocyte antigen (HLA) alleles, particularly HLA-DQ and HLA-DR, which are essential for immune regulation. Environmental triggers, such as viral infections and certain dietary components, can act as catalysts in genetically predisposed individuals. Autoreactive T cells, activated by these environmental factors, infiltrate the pancreatic islets, triggering an immune response against β-cells. This response leads to the release of pro-inflammatory cytokines and chemokines, resulting in the destruction of β-cells [78,79,80]. Several studies have focused on the autoimmune processes involved in pathogenesis, which are described in depth in the literature [16,17]. On the other hand, recent studies indicate that, in addition to this mechanism, altered intestinal permeability, impaired barrier function and the intestinal microbiota can significantly influence the development of diabetes [22,23].

Changes in the intestinal mucosa, such as reduced cytokine production (IL-17A, IL-22 and IL-23A) and shifts in microbiota composition (e.g., a decrease in SFB), have been observed in various mouse models of T1D. These changes do not seem to depend solely on hyperglycemia but are associated with intestinal inflammation and compromised epithelial cell integrity [22,23].

Studies suggest that treating intestinal inflammation could restore the microbiota balance and improve the condition of intestinal epithelial cells, potentially positively influencing T1D. In particular, the use of anti-TNFα therapies, which are common in inflammatory bowel diseases such as Crohn’s disease, has been shown to reduce intestinal inflammation and improve diabetic pathology [23].

Gastrointestinal (GI) symptoms are generally more common in people with diabetes than in the general population, with prevalence rates varying widely. GI complications are commonly nausea, vomiting, abdominal pain, heartburn, dysphagia, constipation, diarrhea and fecal incontinence [81]. Diabetes disrupts intestinal homeostasis through insulin-like growth factor binding protein 3 (IGFBP3). Chronic hyperglycemia and inflammation alter liver function, reducing insulin-like growth factor 1 (IGF-I) and increasing IGFBP3 levels. Elevated IGFBP3 targets intestinal crypts, leading to the apoptosis of intestinal stem cells (ISCs). The remaining ISCs struggle to differentiate into essential cell types, impairing crypt turnover and causing mucosal atrophy, which results in malabsorption. This, in turn, worsens hyperglycemia and inflammation, creating a feedback loop that further disrupts intestinal homeostasis [22,23]. Meanwhile, the role of the enteric nervous system and its neurotransmitters is gaining attention. Damage to this system can lead to specific GI motility disorders, such as diabetic gastroparesis, constipation and diarrhea. These complications increase diabetes-related morbidity and reduce patients’ quality of life [82].

In summary, GI dysfunction significantly increases morbidity in diabetes and creates a vicious cycle of malabsorption, inflammation and impaired GI motility, all of which severely impact patients’ quality of life.

PTPs are crucial regulators of cellular signalling pathways and have also been implicated in the pathogenesis of T1D. PTPs serve as pivotal negative regulators of multiple signalling pathways, influencing both immune responses and glucose metabolism [17,60,83,84,85,86,87,88]. Understanding the specific roles and mechanisms of different PTPs offers valuable insights into their potential as targets for therapeutic strategies aimed at modulating autoimmune responses and managing T1D.

The loss of PTPN2 function seems to be associated with increased intestinal permeability and increased expression of pro-inflammatory mediators, indicating that PTPN2 plays a protective role in maintaining intestinal barrier integrity and regulating the inflammatory response. A polymorphism in the PTPN2 rs2542151 GT or GG genotype is associated with an earlier onset of type 1 diabetes [61]. In addition, PTPN2 rs2542151 has been linked to a higher prevalence of type 2 diabetes (T2DM) and greater severity of non-alcoholic fatty liver disease (NAFLD) [60]. PTPN2 is known for its role as a negative regulator of IFN-γ-mediated pro-inflammatory signalling. IFN-γ increases intestinal permeability and promotes inflammation. This is a process that PTPN2 appears to counteract by limiting the expression of proteins such as claudin-2, which is involved in the formation of pores in the epithelial barrier. These changes may facilitate the access of intestinal antigens to the immune system, contributing to chronic inflammation and insulin resistance [60]. A recent study revealed that reduced expression of the T1D candidate gene PTPN2 exacerbates damage caused by IFN-α and TNF-α. PTPN2 seems to provide protection against both factors, revealing a shared downstream signalling pathway between them. The inhibition of PTPN2 also impairs PDX1 (pancreatic and duodenal homeobox 1) expression and insulin production after TNF-α exposure [62].

PTPN2 is a crucial regulator of the harmful effects of TNF-α in human β-cells, suggesting that patients with PTPN2 risk polymorphisms may benefit from therapies targeting TNF-α. PTPN2 may also influence T1D development through its role in regulating intestinal permeability.

Further research is needed to explore the specific molecular mechanisms underlying T1D progression and, also, what the role of PTPs is in both the development and the progression of the disease. Targeting PTPs could provide new perspectives for therapeutic intervention and disease management [62] (Figure 2 and Table 1).

Interestingly, tissue transglutaminase (TTg), one of the major players in CeD autoimmunity, can modify PTEN, a phosphatase of the PTP family, also increasing the activity binding of the auto-antibodies (IA-2; anti-islet antibodies) participating in the autoimmune response in type 1 diabetes [89]. Moreover, TTg participates in the inflammatory response by activating NFk-β [90]. Considering, also, that in CeD as well as in type 1 diabetes deposits of TTg/anti-TTg are present at the level of the intestinal mucosa [91,92], these two diseases share more common elements than previously thought.

In addition to PTPN2, two other PTPs have only been associated with T2D: PTP-1B and PTPN9 (PTP-MEG2). They are protein tyrosine phosphatases implicated in regulating insulin and leptin signalling pathways, both of which play critical roles in metabolic processes [93,94].

PTP-1B acts as a negative regulator of insulin and leptin receptor pathways. It deactivates insulin and leptin signalling by dephosphorylating key pathway components, impacting downstream effects on glucose metabolism and fatty acid oxidation. As such, PTP-1B inhibition has been identified as a promising strategy for managing type 2 diabetes and related complications, including obesity. Interestingly, metformin can inhibit PTP1B tyrosine phosphatase activity to stimulate the insulin receptor tyrosine kinase [95]. Still, the development of specific PTP-1B inhibitors remains in preliminary stages, with challenges in achieving selectivity and bioavailability due to the conserved structure of PTP active sites [93].

PTPN9 also plays a role in metabolic regulation, with recent studies highlighting its involvement in the pathology of polycythemia vera and its modulation of insulin and ErbB2 receptor signalling. PTPN9 depletion in diabetic mouse liver tissues has been shown to increase insulin sensitivity and normalize blood glucose, indicating that its inhibition could be an effective approach for diabetes treatment. Through an innovative library-based approach, researchers have developed highly selective PTPN9 inhibitors that enhance insulin action and improve glucose homeostasis in cell cultures and diabetic mouse models. This strategy involves targeting peripheral binding pockets near the active site to achieve the necessary specificity and efficacy [94,96].

Together, these findings suggest that selective inhibition of PTP-1B and PTPN9 could offer new therapeutic targets for type 2 diabetes. However, overcoming challenges such as the PTP active site’s high conservation and achieving cell permeability are essential for developing effective small-molecule PTP inhibitors. The research thus lays important groundwork for future PTP-targeted therapies aimed at improving metabolic health and managing diabetes.

## 5. PTP and Celiac Disease

CeD is a multifactorial disease in which environmental triggers combined with genetic predispositions determine its onset. Repeated GWASs (genome-wide association studies) were able to identify 39 non-HLA (human leukocyte antigen) susceptibility loci associated with CeD, and, together, these loci can explain approximately 10–14% of CeD heritability [63]. Combined with the HLA locus, only 50% of CeD heritability is explained. Among these 39 non-HLA loci (Rho involved in CeD), 4 were not directly involved in immunological processes. C1ORF106 (C1 Orfan 106), ARHGAP31 (GTPase activating protein 31), LPP (lipoma-preferred partner) and PTPRK (protein tyrosine phosphatase receptor-type K) play roles in actin rearrangement in the cytoskeleton, in the cell‒cell adhesion process and in the maintenance of intestinal epithelial barrier function [63]. PTPRK also regulates EGFR (epidermal growth factor receptor) signalling through dephosphorylation of catalytic sites and inactivation of the receptor.

The PTPRK gene is located on the long arm of chromosome 6 [64]. PTPRK mRNA is widely expressed, except in the immune cells, skeletal muscles and testes, and it is up-regulated by the transforming growth factor β (TGF-β) signal transduction pathway.

PTPRK is a receptor-type PTP that consists of an extracellular region, a single transmembrane region and two tandem catalytic domains [65]. Many functions are performed by the PTPRK gene [66,67,68,69,70], including the regulation of EGFR activation [66,67,71]. Altered regulation of EGFR has been shown to promote uncontrolled cell proliferation, angiogenesis and metastasis development. Proliferation is also a marker of CeD pathogenesis, mainly through the signalling of EGFR [72]. In biopsies and fibroblasts from patients with CeD, there is a delay in EGFR trafficking in early endocytic vesicles such that the receptor remains activated longer than that in controls [73]. Gliadin further delays the endocytic pathway and can prolong the activation of the EGFR/ERK (extracellular signal-regulated kinase) signalling pathway and cell proliferation in biopsies from controls and patients with CeD [74,75]. Elevated activation of the EGFR/MAPK pathway is also found in the enterocytes of patients with CeD, resulting in alterations in the cytoskeleton. Furthermore, altered expression of genes involved in autophagy and the regulation of the EGFR pathway are also observed in patients with CeD [73].

The effects of PTPRK in regulating the proliferation of enterocytes and keratinocytes have also been studied via biopsies and organoids [18,76,77]. PTPRK phosphatase expression levels are reduced in biopsies from patients with a gluten-containing diet (GCD) and from potential patients with CeD (Pot-CeD), whereas EGFR phosphorylation, ERK and cell proliferation levels are increased compared with those in biopsies from controls (CTRs). These findings suggest that the reduction in PTPRK protein expression levels is independent of intestinal villous atrophy [18].

Another study found a region on chromosome 6 through a GWAS that includes two genes, the THEMIS gene, which codes for a molecule involved in the selection of thymocytes, and the PTPRK gene. Both genes perform immuno-related functions and are involved in the pathogenesis of CeD. Gene expression analysis of duodenal biopsies from patients with CeD revealed higher mRNA levels in the THEMIS gene than those in samples taken from the same patients after gluten-free diet (GFD) treatment or from CTRs. These findings point to the correlation between low PTPRK expression levels in the CeD and increased permeability of the intestinal barrier, and PTPRK inhibition may promote enterocyte hyper-proliferation in intestinal crypts. Moreover, compared with CTR patients, patients with CeD have similar mRNA levels, indicating that adhesion to a GFD is sufficient to restore the transcriptional activity of the THEMIS and PTPRK genes to normal levels [77].

Altered function of the intestinal epithelium plays an important role in the pathogenesis of inflammatory diseases [97]. Intestinal organoids, derived from intestinal stem cells, recreate the crypt/villus axis in vitro, allowing the role of the intestinal epithelium in various pathologies to be studied [98,99,100,101,102]. CeD intestinal organoids reproduce the low-grade inflammation of CeD intestinal biopsies [103]. In CeD organoids, low PTPRK levels are associated with increased EGFR/ERK phosphorylation and proliferation. Nanayakkara et al. conducted functional experiments on intestinal organoids from patients with CeD and controls to study the role of PTPRK in intestinal epithelial cell proliferation. In control organoids, the silencing of PTPRK induces the increase in the phosphorylation of EGFR and ERK and proliferation. On the other hand, overexpressing PTPRK in CeD organoids was able to reduce the phosphorylation of EGFR and ERK and proliferation to normal levels. PTPRK expression in intestinal organoids from patients with CeD and controls can modulate the phosphorylation of EGFR and ERK and proliferation [18] (Figure 2 and Figure 3 and Table 1).

The discovery of the role of PTPRK in CeD organoids is exemplary of how organoids can be used to understand specific features of intestinal diseases. In fact, they can allow for the study of an important feature of CeD, such as proliferation, which is a hallmark of CeD, and also the role of PTPs in the pathogenesis of CeD. This shifts the attention from T-cell response to the intestinal epithelium in this disease.

## 6. Conclusions

In conclusion, the involvement of PTP phosphatases in diseases such as IBD, T1D and CeD implicates the nodal role of the signalling mediated by phospho-tyrosines in these diseases. PTPs are signalling molecules that operate in almost all cells of the body and are able to regulate a variety of cellular processes, such as cell growth, differentiation, the cell cycle and oncogenic transformation. The generation of different diseases is likely due to the fragility of specific cell tissues induced by a combination of exogenous and endogenous factors. Moreover, these data point to an emerging role of the intestinal epithelium not only in inflammatory chronic intestinal diseases, including IBD and CeD, but also in T1D, where PTP alterations have been found both at the level of pancreatic β-cells and at the level of intestinal epithelial cells. Taken together, these data link IBD, T1D and CeD, which were previously regarded as unrelated and mediated by independent pathogenetic events. Moreover, although previously considered “undraggable”, PTPs are now actively studied to develop new drugs. Considering also the availability of organoids as models to test new drugs in personalized way, it is very likely that, soon, these proteins will be the targets of useful drugs.

## Figures and Tables

**Figure 1 cells-13-01981-f001:**
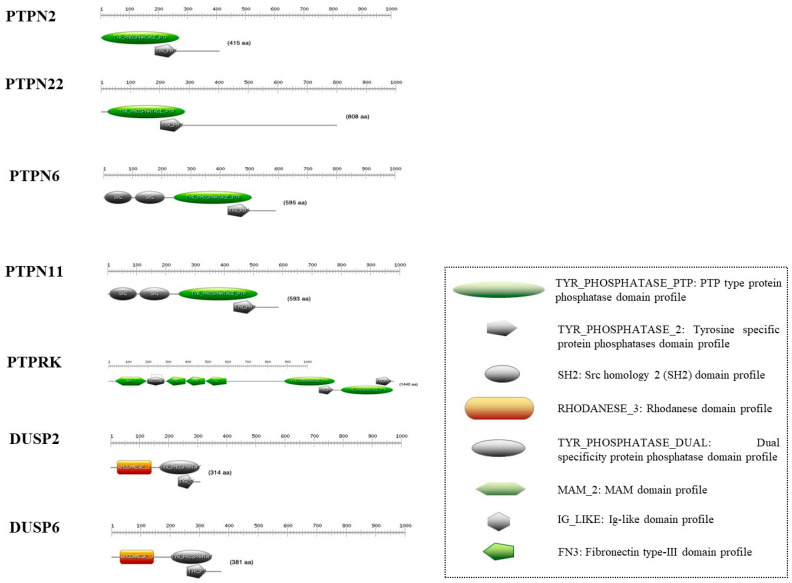
Graphical representation of the PTP domain profile.

**Figure 2 cells-13-01981-f002:**
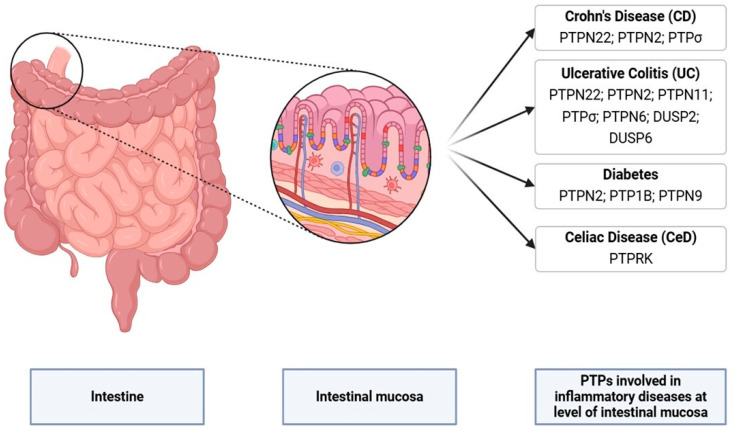
Schematic representation of PTPs (PTPN22; PTPN2; PTPσ; PTPN11; PTPN6; DUSP2; DUSP6; PTPRK) involved in Crohn’s disease, ulcerative colitis, diabetes and celiac disease at level of intestinal mucosa.

**Figure 3 cells-13-01981-f003:**
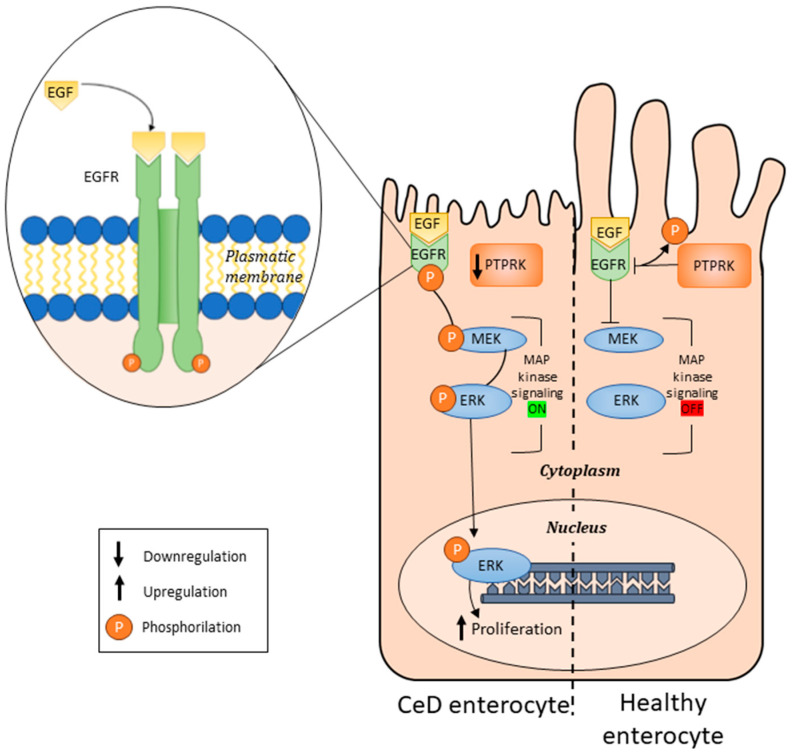
Schematic representation of CeD (**left**) and healthy (**right**) enterocytes with a description of the role of PTPRK in the regulation of the EGF pathway. In CeD enterocytes, with respect to healthy enterocytes, the downregulation of PTPRK leads to the upregulation of EGFR, and MAP-kinase signalling is always switched ON, increasing proliferation.

## Data Availability

Not applicable.

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
