# Peer review of "Role of Protein Tyrosine Phosphatases in Inflammatory Bowel Disease, Celiac Disease and Diabetes: Focus on the Intestinal Mucosa"

_cells, 2024, doi:10.3390/cells13231981_

Round 1

Reviewer 1 Report

Comments and Suggestions for Authors

Dear Authors,

The manuscript presents a theme of significant scientific relevance. However, I have only a few comments.

I have listed my comments below:

  1. You should include the graphical abstract, which could increase the article's visibility.
  2. I recommend that the authors, including the software used to create the figures.
  3. Why did the authors not address transglutaminase in the manuscript?
  4. Do the authors not consider addressing metformin and gastrointestinal parameters (protein phosphatases) in diabetic patients?

Author Response

Comments 1: You should include the graphical abstract, which could increase the article's visibility?

Response 1: We thank referee for the comments. A graphical abstract has been done and added as a PDF to our point to point response to the referees.

Comments 2: I recommend that the authors, including the software used to create the figures.

Response 2: In accord with the referee comment we have added the following paragraph to the Material and Methods section, Lines 108-110:

“The software used to create the figures were Clustal, Microsoft Power Point and BioRender. In particular, Clustal was used to create Figure 1; BioRender was used for Figure 2 and Microsoft Power Point for the graphical abstract Figure 3.”

Comments 3: Why did the authors not address transglutaminase in the manuscript?

To clarify this point, we have added the following paragraph to the section “PTP and Diabetes”, Lines 289-295:

Response 2: Interestingly, tissue trans glutaminase (TTg), one of the major players in CeD autoimmunity can modify PTP-N a phosphatase of the PTP family, increasing the activity binding of the auto-antibodies (IA-2 Tyrosine-phosphatase 2 protein) participating to the autoimmune response also in Type 1 diabetes (74). Moreover, TTg participate to the inflammatory response by activating NFkB (75). Considering, also, that in CeD as well as in Type 1 diabetes deposits of TTg/anti-TTg are present at the level of the intestinal mucosa (76,77) these two diseases share several common elements than previously thought.

Comments 4: Do the authors not consider addressing metformin and gastrointestinal parameters (protein phosphatases) in diabetic patients?

Response 4:

  1. Metformin is the main therapeutic choice in Type 2 diabetes, although it can be used in severe cases of Type 1 diabetes with insulin resistance. Metformin can have several functions, among them, it can inhibit PTP1B tyrosine phosphatase activity to stimulate the insulin receptor tyrosine kinase (66). To clarify this point we have added the following paragraph to the section “PTP and Diabetes”, Lines 304-305:

“Interestingly, metformin can can inhibits PTP1B tyrosine phosphatase activity to stimulate the insulin receptor tyrosine kinase (80).”

  1. We thank referee 1 for pointing to the role of gastrointestinal parameters in T1D. To clarify this point, we have added the following paragraph to the section “PTP and Diabetes”, Lines 240-257:

“Gastrointestinal (GI) symptoms are generally more common in people with diabetes than in the general population, with prevalence rates varying widely. GI complications are commonly nausea, vomiting, abdominal pain, heartburn, dysphagia, constipation, diarrhea, and fecal incontinence (63). Diabetes disrupts intestinal homeostasis through insulin-like growth factor binding protein 3 (IGFBP3). Chronic hyperglycemia and inflammation alter liver function, reducing insulin-like growth factor 1 (IGF-I) and increasing IGFBP3 levels. Elevated IGFBP3 targets intestinal crypts, leading to the apoptosis of intestinal stem cells (ISCs). The remaining ISCs struggle to differentiate into essential cell types, impairing crypt turnover and causing mucosal atrophy, which results in malabsorption. This, in turn, worsens hyperglycemia and inflammation, creating a feedback loop that further disrupts intestinal homeostasis (22,23). Meanwhile, the role of the enteric nervous system and its neurotransmitters is gaining attention. Damage to this system can lead to specific GI motility disorders, such as diabetic gastroparesis, constipation, and diarrhea. These complications increase diabetes-related morbidity and reduce patients' quality of life (64).

In summary, GI dysfunction significantly increases morbidity in diabetes and creates a vicious cycle of malabsorption, inflammation, and impaired GI motility, all of which severely impact patients' quality of life.”

Reviewer 2 Report

Comments and Suggestions for Authors

In their review, the authors summarise the role of the various protein tyrosine phosphatases in three different diseases: inflammatory bowel disease, coeliac disease and type 1 diabetes. While the pathological changes to the enzymes are well summarised, I would expand the general information on the protein class. That is, are there general differences in the presence of the proteins in different cells and tissues? The authors mention the two main functions of proteins: Immune response and signalling pathways. This is rather vague and should be described in much more detail. There is also an illustration. How do the main functions change under pathological conditions? Are hereditary diseases known for the proteins? Is it possible to generalise overall changes independently of the disease in particular? 

In their review, the authors summarise the role of the various protein tyrosine phosphatases in three different diseases: inflammatory bowel disease, coeliac disease and type 1 diabetes. To this aim, the literature was analysed with regard to the presence and possible role of the enzymes. The abundance of literature shows that this is a relevant topic. I could not find a summary of this kind with the context to this extent from recent years. Overall, the proteins offer an interesting approach to understanding the disease and possible therapeutic approaches. I have the following suggestions for improving the review: While the pathological changes to the enzymes are well summarised, I would expand the general information on the protein class. How different are the individual representatives, do they complement each other functionally and or are the functionally overlapping? Are there general differences in the presence of the proteins in different cells and tissues? The authors mention the two main functions of proteins: Immune response and signalling pathways. This is rather vague and should be described in much more detail. There is also an illustration. How do the main functions change under pathological conditions? Are hereditary diseases known for the proteins? Is it possible to generalize overall changes independently of the disease in particular? Furthermore, the authors should highlight open questions about the protein class? In which direction would further research be useful? In other words, provide more impetus and suggestions to understand the role and above all the significance of the enzymes for the pathogenesis of the diseases and to substantiate possible therapeutic approaches. The references are appropriate, and I have provided additional illustrations above, for example to show the function of the enzymes or the presence of the individual representatives.

Author Response

Comments 1: While the pathological changes to the enzymes are well summarised, I would expand the general information on the protein class.

Response 1: In agreement with referee we have clarified this point by erasing the following paragraph to the “Introduction” section, Lines 60-63

“Class I includes both classical PTP phosphatases that add phosphorylated tyrosine residues and dual-specific phosphatases (DUSPs) that target phosphorylated tyrosine, serine, and threonine residues.”

And adding Lines 59-65

“Human PTPs are classified into four categories depending on the substrate specificity and amino acid sequence. Class 1 consists of the classical PTPs, including the receptor-type protein tyrosine phosphatases (RPTPs) and non-receptor-type protein tyrosine phosphatases (nRPTPs). Class 2 comprises the dual-specificity phosphatases (DSPs) that target phosphorylated tyrosine, serine, and threonine residues. The low molecular weight phosphatases (LMW-PTPs) belong to class 3 and the CDC25 class phosphatases form class 4 (10).”

Comments 2: That is, are there general differences in the presence of the proteins in different

cells and tissues?

Response 2: To clarify this point we have added the following paragraph to the Introduction section, Lines: 65-68

“At the cellular level the PTP proteins are mainly cytosolic, nucleo-cytosolic and receptor proteins. Although at the tissue level they are expressed mainly in the developing embryo, the immune and the nervous system, they are also expressed at the level of the intestine (11)”

Comments 3: The authors mention the two main functions of proteins: Immune response and

signalling pathways. This is rather vague and should be described in much more detail. There is also

an illustration.

Response 3: We agree with referee  that the functions of the PTP proteins have not been addressed properly. For this reason, we have erased in the Introduction section the following paragraph Lines 52-54:

“PTPs are also classified for their main functions in the immune response (PTPN22, PTPN6, DUSP2, DUSP6, and PTPN11) and signalling pathways (PTPN2, PTPN6, PTPN11, PTPσ, DUSP6, DUSP2, and PTPRK)”

And added Lines 48-51:

PTP proteins play a nodal role in the regulation of signals mediated by phosphatases. “Therefore, PTPs are signalling molecules that regulate a variety of cellular processes, such as cell growth, differentiation, the cell cycle, oncogenic transformation, embryonic development, immune response, inflammation, proliferation, differentiation, cell adhesion and autophagy (4-7).”

 Comments 4: How do the main functions change under pathological conditions?

Response 4: In agreement with referee we have clarified this point by adding the following paragraph to the “Introduction” section Lanes 71-73:

“The importance of tyrosine phosphorylation in normal cell physiology is highlighted by the fact that many human diseases are the result of aberrant protein tyrosine kinase (PTK) or protein tyrosine phosphatase (PTP) function (12).”

Comments 5: Are hereditary diseases known for the proteins?

Response 5: In agreement with referee to clarify this point we have added the following paragraph to the “Introduction” section, Lines 73-76:

“PTPs mutations that can cause hereditary disease phenotypes have been described to date for T-cell Leukemia, Rheumatoid arthritis, Hashimoto thyroiditis, Addison Disease, Noonan and Leopard syndrome and others (13).”

Comments 6: Is it possible to generalise overall changes independently of the disease in particular?

Response 6: In agreement with the referee to clarify this point we have added the following paragraph to the Introduction Section, Lines 76-77

“Interestingly both loss-of-function and gain-of-function mutations in these genes may underlie the reported disease”

Comments 7: How different are the individual representatives, do they complement each other functionally and or are the functionally overlapping?

Response 7: In agreement with the referee we have clarified this point adding to the “Introduction” Section, Lines 37-48

The kinases and phosphatases have complementary functions in regulating cell signalling with the kinases controlling the amplitude of a signalling response and the phosphatases controlling the rate and duration of the response. PTPs are a super family with 100 circa different genes and the kinases are 90 circa, indicating the same level of complexity for the two families (2). Moreover, they reach a higher level of complexity considering post-transcriptional modifications, alternative promoters or alternative splicing. Such diversity, on one side, makes a neat classification hard, on the other side points to a wide range of biological functions not completely clarified. Understanding the “phosphatome”, is nowadays a real challenge to understand the specificity and roles of all phosphatases. In fact, it is not possible to generalise the function of each phosphatase: some of them are very selective whereas others are more promiscuous (3).

Comments 8: Are there general differences in the presence of the proteins in different cells and tissues?

Response 8: See response 2 to comment 2.

Comments 9: The authors mention the two main functions of proteins: Immune response and signalling pathways. This is rather vague and should be described in much more detail. There is also an illustration.

Response 9: See response 3 to comment 3

Comments10: How do the main functions change under pathological conditions?

Response 10: See response 4 to comment 4

Comments 11: Are hereditary diseases known for the proteins?

Response 11: See response 5 to comment 5

Comments 12: Is it possible to generalize overall changes independently of the disease in particular?

Response 12: See response 6 to comment 6

Comments 13: Furthermore, the authors should highlight open questions about the protein class?

Response 13: In agreement with the referee 2 we have added the following paragraph in the “Introduction” section Lines: 40-45

“PTPs are a super family with 100 circa different genes that compared to the 90 circa kinases indicate the same level of complexity for the two families (2). Moreover, they reach a higher level of complexity considering post-transcriptional modifications, alternative promoters or alternative splicing. Such diversity on one side makes a neat classification hard, on the other side points to a wide range of biological functions to not completely clarified.”

Comments 14: In which direction would further research be useful?

Response 14:  In agreement with the referee comment we have added the following paragraph to the introduction section Lines: 45-48

“Understanding the “phosphatome”, is nowadays a real challenge to understand the specificity and roles of all phosphatases. In fact, it is not possible to generalise the function of each phosphatase: some of them are very selective whereas others are more promiscuous (3).

and the conclusion section lines: 411-414

 Moreover, although previously considered “undraggable”, PTPs are now actively studied to develop new drugs. Considering, also the availability of organoids as models to test new drugs in personalized way, very likely, soon these proteins will be targets of useful drugs.”

Comments 15: In other words, provide more impetus and suggestions to understand the role and above all the significance of the enzymes for the pathogenesis of the diseases and to substantiate possible therapeutic approaches.

Response 15: See response14

Comments 16: I have provided additional illustrations above, for example to show the function of the enzymes or the presence of the individual representatives.

Response 16: We thank referee for providing additional figures, but we have not received them, but now we have provided a graphical abstract that shown the function of the cited PTPs.

Reviewer 3 Report

Comments and Suggestions for Authors

The manuscript entitled "Role of Protein Tyrosine Phosphatase in Inflammatory Bowel Disease, Celiac Disease and Diabetes: focus on the intestinal mucosa" written by C. Bellomo et al aims to review the literature concerning the role of several PTPs —PTPN22, PTPN2, PTPN6, PTPN11, DUSP2, DUSP6 and PTPRK— at the level of the intestinal mucosa in inflammatory bowel disease (IBD), celiac disease (CeD), and type 1 diabetes (T1D) in both in vitro and in vivo models. 

The manuscript is well-written and structured and responds to important questions addressed in our days.

Completing this study with the literature data about the studied PTPs in type 2 diabetes will add value and consistently improve the proposed review.

Author Response

Comments 1: Completing this study with the literature data about the studied PTPs in type 2 diabetes will add value and consistently improve the proposed review.

Response 1: We thank referee for pointing out this issue. In agreement with referee we have now added a new paragraph, as follows, to the “PTPs and diabetes” section, Lines 296-323 in which we describe the PTP studied in type 2 diabetes with the relative bibliography.

“In addition to PTPN2, two other PTPs have been associated only with T2D: PTP-1B and PTPN9 (PTP-MEG2). They are protein tyrosine phosphatases implicated in regulating insulin and leptin signaling pathways, both of which play critical roles in metabolic processes (78,79)

PTP-1B acts as a negative regulator of insulin and leptin receptor pathways. It deactivates insulin and leptin signaling by dephosphorylating key pathway components, impacting downstream effects on glucose metabolism and fatty acid oxidation. As such, PTP-1B inhibition has been identified as a promising strategy for managing type 2 diabetes and related complications, including obesity. Interestingly, metformin can can inhibits PTP1B tyrosine phosphatase activity to stimulate the insulin receptor tyrosine kinase (80). Still, the development of specific PTP-1B inhibitors remains in preliminary stages, with challenges in achieving selectivity and bioavailability due to the conserved structure of PTP active sites (78).

PTPN9 also plays a role in metabolic regulation, with recent studies highlighting its involvement in the pathology of polycythemia vera and its modulation of insulin and ErbB2 receptor signaling. PTP-MEG2 depletion in diabetic mouse liver tissues has been shown to increase insulin sensitivity and normalize blood glucose, indicating that its inhibition could be an effective approach for diabetes treatment. Through an innovative library-based approach, researchers have developed highly selective PTP-MEG2 inhibitors that enhance insulin action and improve glucose homeostasis in cell cultures and diabetic mouse models. This strategy involved targeting peripheral binding pockets near the active site to achieve the necessary specificity and efficacy (79,81).

Together, these findings suggest that selective inhibition of PTP-1B and PTP-MEG2 could offer new therapeutic targets for type 2 diabetes. However, overcoming challenges such as the PTP active site's high conservation and achieving cell permeability are essential for developing effective small-molecule PTP inhibitors. The research thus lays important groundwork for future PTP-targeted therapies aimed at improving metabolic health and managing diabetes.”

Reviewer 4 Report

Comments and Suggestions for Authors

The topic of the manuscript “Role of Protein Tyrosine Phosphatase in Inflammatory Bowel Disease, Celiac Disease and Diabetes: focus on the intestinal mucosa” discussing protein tyrosine phosphatases (PTPs) in inflammatory diseases such as IBD, celiac disease, and diabetes is undoubtedly relevant and important. However, the delivery dampens interest. The content is presented in a dry manner, without emphasis on broader implications or connections to practical applications. Additionally, the Materials and Methods section is especially vague and confusing, for example, in the statement about reviewing literature: "and 28/83 from the last 5 years and only are between 1993 and 1998". The significance of these figures is unclear, and the phrasing disrupts reader engagement.

Furthermore, the manuscript lacks a clear focus in places. It attempts to cover a wide range of PTPs across three diseases but doesn't consistently maintain depth in its analysis of each. There are moments where the text diverges into detailed, jargon-heavy descriptions without providing adequate background for each element (e.g., listing various PTPs without fully explaining their relevance). The Introduction doesn't adequately set the stage for the reader, leaving gaps in understanding that weaken the focus of the manuscript.

Overall, the manuscript suffers from significant readability issues. The writing is often dense, technical, and lacks smooth transitions between ideas, making it difficult to follow. For example, in the Introduction, the text feels like a series of disjointed bullet points rather than a coherent narrative. Sentences don't flow well, and there is little context provided for terms or concepts, which would make it hard for readers who are not experts in the field to engage with the content.

Recommendations:

  1. Title: The title should correctly reflect the subject matter by specifying "phosphatases" rather than "phosphatase."
  2. Text: Needs substantial reworking to establish clear connections between the concepts and terms used. For example, the Introduction should start with a broad overview before diving into specific mechanisms.
  3. Materials and Methods: Clarify how the literature was selected and what is meant by the 1993-1998 references. It reads as incoherent, and clearer methodology is essential for a review.
  4. Overall Structure: The manuscript would benefit from better transitions and explanations of why certain data are significant. Adding summaries or reflections at the end of sections could help tie ideas together more cohesively.

Comments on the Quality of English Language

The quality of the English language is adequate, but certain sections would benefit from revisions to improve clarity and fluency.

Author Response

Comment 1: Title: The title should correctly reflect the subject matter by specifying "phosphatases" rather than "phosphatase."

Response 1: We thank the referee for pointing this mistake out, the Title has been corrected as follows:

Role of Protein Tyrosine Phosphatases in Inflammatory Bowel Disease, Celiac Disease and Diabetes: focus on the intestinal mucosa”

Comments 2: Text: Needs substantial reworking to establish clear connections between the concepts and terms used. For example, the Introduction should start with a broad overview before diving into specific mechanisms.

Response 2: In agreement with the referee comments, we have done several changes to the text as follows

  • We have completely rewritten the “Introduction “section (Lines 37-51,59-68,71-77) by adding new paragraph addressing a broad overview, reducing the specific mechanisms, and traying to establish clear connections between the concepts and terms used.
  • We have simplified the “detailed, jargon-heavy descriptions” as follows

  1. Section “PTPs and Inflammatory Bowel Disease” we have modified the text as follows

“Gene association study demonstrated that SNPs that generating loss-of-function variant in PTPN22 were associated respectively with CD and UC (4-6)” eliminating the detailed description of the mutations.

  1. In the section “PTPs and Inflammatory Bowel Disease” we have erased the following paragraph Lines 124-131
  • In addition, PTPN2 is involved in autophagy regulation and inflammasome activation (7). PTPN2 was shown to regulate inflammasome activation and IL-1B secretion by limiting the phosphorylation of the adaptor molecule apoptosis-associated speck-like protein containing a CARD (ASC), which is required for inflammasome assembly. In mice, the loss of PTPN2 in myeloid cells results in increased susceptibility to DSS-induced colitis due to increased inflammasome assembly and elevated IL1B production at the level of the intestine (8) (Fig. 2 and Tab.1).

And added to the same paragraph 148-149

  • “As well as PTPN22 also PTPN2 was shown to regulate inflammasome activation and IL-1B secretion.”(40)

And, also, modified the same paragraph eliminating too detailed descriptions:

  • “Clinically relevant loss-of-function mutations in the PTPN2 gene contribute to the etiopathogenesis of chronic inflammatory intestinal disease, at least in part through dysregulation of epithelial barrier properties. As well as PTPN22 also PTPN2 was shown to regulate inflammasome activation, IL-1B secretion and autophagy regulation (7) Consistent with these findings, PTPN2 levels were increased in the intestinal epithelium in active CD patients. (21, 42, 43). In fact, PTPN2 protect the intestinal barrier by restricting the capacity of INF gamma to increase epithelial permeability and TNFα-induced signalling and cytokine secretion in human intestinal epithelial cells (21,42-44)

And the Paragraph “PTPN11”, Lines 155-167 has been modified as follows:

  • PTPN11 was upregulated in both UC patients and colitis model mice (45). It is Src homology 2 (SH2) domain-containing tyrosine phosphatase-2 (SHP2, encoded by the gene PTPN11) (46). It is a cytoplasmic protein tyrosine phosphatase that functions as a positive regulator of the Ras–mitogen-activated protein kinase (MAPK) signalling pathway downstream of various growth factors and cytokines, it is thought to contributing to the regulation of cell proliferation and differentiation. On the other hand, Yamashita et al demonstrated that SHP2 PTPN11 lacking mice specifically in IECs developed severe colitis, alterations of intestinal mucosa with numbers of absorptive enterocytes and goblet cells markedly reduced in the small intestine and colon respect to controls(47). Abnormal development of the intestinal epithelium and disrupted host–microbiota equilibrium also was described in PTPN11 deficient mice. (49, 50) SHP2 is necessary for the homeostasis of IECs, particularly absorptive enterocytes and goblet cells, as well as for protection against colitis, as evaluated by a previous study in the small intestine and the development of intestinal organoids from isolated crypts in mice. Moreover, Xiao et al. demonstrated that Shp2 expressed in the monocyte/macrophage lineage serves as a proinflammatory factor in the intestine (48). In contrast, mice with SHP2 deficiency in intestinal epithelial cells exhibit abnormal development of the intestinal epithelium and disrupted host–microbiota equilibrium (49, 50). However, the precise role of PTPN11 in the regulation of gastrointestinal stem cell and progenitor functions, especially in vivo, has remained unclear (Fig. 2 and Tab.1).”

  • PTPsigma paragraph has been modified as follows, Lines 168-175:

PTPσ, (Protein Tyrosine Phosphatase Receptor S), consists of a cell adhesion molecule-like domain containing three immunoglobulin (Ig)-like repeats and three to eight fibronectin type III repeats, a transmembrane domain, and a cytosolic region with two PTPase domains, of which the first (D1) is catalytically active.

  • PTPN6 paragraph has been modified as follows, Lines 176  PTPN6, previously known as SH-PTP1, is an Src homology region 2 domains–containing tyrosine phosphatase that is predominantly expressed in hematopoietic and epithelial cells.”

  • DUSP2 paragraph has been modified as follows, Lines 183-189: “DUSP2(also known as Procaspase-Activating Compound 1− PAC-1) was first identified as a mitogen-inducible gene in human T cells. DUSP2 is a nuclear protein phosphatase that is regulated by the dephosphorylation of the extracellular signal-regulated kinase 1/2  (ERK1/2) and p38 MAPKs. DUSP2 is a positive mediator of inflammation (55).  DUSP2−/− mice exhibit more severe disease, increased mucosal hyperaemia, increased colonic ulceration, and high levels of proinflammatory cytokines, including IL-6, IL-17, Tumor necrosis factor α (TNFα), and IL-1β. In a study of UC patients, the level of blood DUSP2 mRNA expression was reduced. in their blood because of the methylation of CpG islands (56) (Fig. 2 and Tab.1).”

DUSP6 paragraph has been modified as follows, Lines 190-199:

  • DUSP6 is a cytoplasmic phosphatase specific to the ERK protein kinase (58). The signal transduction pathway regulated by ERK is responsible for the regulation of multiple cellular responses. DUSP6KO mice have shown that ERK induces CD4+ T cell polarization, altered differentiation of T-helper 1 (TH1), TH2, TH17 cells. DUSP6-deficient mice presented not only increased CD4+ T cell activation and an increased rate of cell proliferation induced by ERK but also and an increased rate of cell death. This depletion also causes severe inflammation in mouse models of IBD, with as confirmed by histological examination. It shows prominent intestinal epithelial hyperplasia, depletion of caliciform cells and infiltration of mononuclear cells. Furthermore, colon explants secrete increased levels of TNF-α and Interferon-γ (IFN-γ), whereas IL-17α levels are reduced. In these mice, ERK inhibition significantly reduces colitis. Additionally, Beaudry K. et al., by ex vivo DUSP6 KO organoid cultures, demonstrated that loss of DUSP6 caused an increase in crypt depth and epithelial cell proliferation (58) (Fig. 2 and Tab.1).

  Comments 3: Materials and Methods: Clarify how the literature was selected and what is meant by the 1993-1998 references. It reads as incoherent, and clearer methodology is essential for a review.

Response 3: We thank referee for highlight this point, in fact a typing mistake made the “Material and methods” not readable. We have now rewritten this section as follows Lines 104-107: 

“We selected the most recent literature on IBD, T1D, and CeD, which are correlated with/associated with PTP proteins. Citing 72/103 from the last 20 years, 51/103 from the last 10 years and 36/103 from the last 5 years and only 3 are between 1993 and 1998 as they are the original references for the PTP described.”

Comments 4: Overall Structure: The manuscript would benefit from better transitions and explanations of why certain data are significant. Adding summaries or reflections at the end of sections could help tie ideas together more cohesively.

Response 4: In agreement with referee comments we have added al conclusive paragraph to the sections:

  • “PTPs and Inflammatory Bowel Disease” LINES 201-208:

“In conclusion, PTPs have a nodal role in IBD pathogenesis. It is to note that they play an important role in the intestinal epithelium, mostly regulating its homeostasis. This indicating that the intestinal epithelium compartment could be the key both to the understanding the of the molecular pathways responsible for the mucosal alterations, and to find new targets for therapeutical intervention. In this context, it is important to highlight the role of intestinal organoids, derived from staminal cells that represent the intestinal epithelium not only of a specific disease, but also of a specific patient, opening the door of precision medicine in these diseases. “

  • “PTPs and Diabetes”,
  1. LINES 285-288: “Further research is needed to explore the specific molecular mechanisms underlying T1D progression and, also what is the role pf PTPs in both the development and the progression of the disease. Targeting PTPs could provide new perspectives for therapeutic intervention and disease management (72) (Fig. 2 and Tab.1).”
  2. LINES 318-323: “Together, these findings suggest that selective inhibition of PTP-1B and PTPN9 could offer new therapeutic targets for type 2 diabetes. However, overcoming challenges such as the PTP active site's high conservation and achieving cell permeability are essential for developing effective small-molecule PTP inhibitors. The research thus lays important groundwork for future PTP-targeted therapies aimed at improving metabolic health and managing diabetes.”
  3. LINES 386-390: “The discovery of the role of PTPRK in CeD organoids is exemplary not only of how the intestinal epithelium can regulate an important feature of CeD such as proliferation, a hallmark of CeD, but also the role of PTPs in the pathogenesis of this disease. This shifting the attention from T-cells response to the intestinal epithelium in this disease. “

Round 2

Reviewer 4 Report

Comments and Suggestions for Authors

The manuscript has been significantly improved and is now highly comprehensive, engaging to read, and timely. It will undoubtedly attract broad interest among the readers of Cells and beyond.

I have only one remark: citations 4–6 are missing (page 2, line 52).